# Effects of Chronic Kidney Disease and Uremic Toxins on Extracellular Vesicle Biology

**DOI:** 10.3390/toxins12120811

**Published:** 2020-12-21

**Authors:** Linda Yaker, Saïd Kamel, Jérôme Ausseil, Agnès Boullier

**Affiliations:** 1MP3CV-UR7517, CURS-Université de Picardie Jules Verne, Avenue de la Croix Jourdain, F-80054 Amiens, France; linda.yaker@hotmail.fr (L.Y.); said.kamel@u-picardie.fr (S.K.); 2Laboratoire de Biochimie CHU Amiens-Picardie, Avenue de la Croix Jourdain, F-80054 Amiens, France; 3INSERM UMR1043, CNRS UMR5282, University of Toulouse III, F-31024 Toulouse, France; ausseil.j@chu-toulouse.fr; 4CHU PURPAN—Institut Fédératif de Biologie, Laboratoire de Biochimie, Avenue de Grande Bretagne, F-31059 Toulouse, France

**Keywords:** extracellular vesicles, chronic kidney disease, vascular calcification, uremic toxins

## Abstract

Vascular calcification (VC) is a cardiovascular complication associated with a high mortality rate, especially in patients with diabetes, atherosclerosis or chronic kidney disease (CKD). In CKD patients, VC is associated with the accumulation of uremic toxins, such as indoxyl sulphate or inorganic phosphate, which can have a major impact in vascular remodeling. During VC, vascular smooth muscle cells (VSMCs) undergo an osteogenic switch and secrete extracellular vesicles (EVs) that are heterogeneous in terms of their origin and composition. Under physiological conditions, EVs are involved in cell-cell communication and the maintenance of cellular homeostasis. They contain high levels of calcification inhibitors, such as fetuin-A and matrix Gla protein. Under pathological conditions (and particularly in the presence of uremic toxins), the secreted EVs acquire a pro-calcifying profile and thereby act as nucleating foci for the crystallization of hydroxyapatite and the propagation of calcification. Here, we review the most recent findings on the EVs’ pathophysiological role in VC, the impact of uremic toxins on EV biogenesis and functions, the use of EVs as diagnostic biomarkers and the EVs’ therapeutic potential in CKD.

## 1. Introduction

Over the last few years, preclinical and clinical studies have emphasized the role of extracellular vesicles (EVs) in human diseases. These particles are delimited by a lipid bilayer and are released by almost all cell types and in all organisms. EVs appear to have biological effects in various pathophysiological situations and especially in renal disease [1,2]. In human organs, EVs can interact with cells and prompt the release of many different molecules, such as proteins, lipids and nucleic acids, that, in turn, regulate various cell signaling pathways [2]. Moreover, EVs are present in the urine and the blood and therefore can be used as potential diagnostic biomarkers in human diseases, such as chronic kidney disease (CKD, also known as chronic renal failure) [3]. CKD is now a major public health issue. The overall prevalence of CKD is rising worldwide [4]. Indeed, a recent review and meta-analysis found a global CKD prevalence of 13.4% (95% confidence interval: 11.7–15.1%) [4]. CKD affects kidney function and/or structure and has major health consequences [5]. Risk factors for CKD include age, cardiovascular disorders (such as hypertension), diabetes mellitus, obesity and smoking [6]. Vascular calcification (VC) is a strong predictor of outcomes in CKD and, it is associated with high morbidity and mortality rates in CKD patients [7]. Indeed, the prevalence of VC is higher in CKD patients than in the general population [7]. In a CKD setting, VC leads to a decrease in blood vessel elasticity, an increase in blood pressure and therefore to serious cardiovascular complications. VC is a complex pathological process caused by an imbalance between activators of calcification (phosphorus and calcium) and inhibitors of calcification (matrix Gla protein, fetuin-A, osteoprotegerin and osteopontin) [8]. This dysregulation prompts vascular smooth muscle cells (VSMCs) found in the media layer of blood vessels to turn into osteoblast-like cells; this switching may have an important role in the onset of VC [9]. However, other cells (such as osteoclast-like cells, endothelial progenitor cells and Gli1+ mesenchymal stem cells (MSCs)) are also involved in VC [10]. It is now well established that several different mechanisms (such as inflammation and oxidative stress) enhance VC in CKD patients [10]. Moreover, the high prevalence of VC in CKD is associated with the accumulation of uremic toxins that can have a major impact on vascular remodeling [11]. Indeed, the uremic syndrome in patients with advanced CKD can be responsible for many different symptoms, including anemia, bone-mineral disorders and hypertension. Here, we review the links between EVs, VC and uremic toxins in renal diseases in general and in CKD in particular.

## 2. The Biology of Extracellular Vesicles

### 2.1. Classification of EVs

In recent years, studies of the role of EVs in human disease have shown that these vesicles are involved in several physiological pathways. Firstly, EVs are involved in cell–cell communication because they deliver various bioactive molecules and components to recipient cells [1]. In fact, the vesicles carry proteins, lipids, nucleic acids and other metabolites that can have many different cellular and supracellular effects [1]. EVs are highly heterogeneous but can be divided into three main categories: exosomes (30–150 nm in diameter), microvesicles (also referred to as microparticles or ectosomes: 100–1000 nm) and apoptotic vesicles (50–5000 nm) [12]. The three subgroups differ with regard to their mode of biogenesis (Figure 1).

### 2.2. Biogenesis of EVs

#### 2.2.1. The Biogenesis of Exosomes

Exosomes are produced from a complex endosomal system involving two different pathways, one of which depends on a protein complex called the endosomal sorting complex, which is required for transport (ESCRT) [13]. Typically, the budding of late endosomes enables the production of intraluminal vesicles within multivesicular endosomes. These vesicles then fuse with the plasma membrane to give secreted exosomes [13].

In the ESCRT-dependent pathway, four protein complexes (ESCRT-0, I, II, III) bind to the hexameric AAA ATPase Vps4 complex, which provides the energy for exosome biogenesis [13]. The intraluminal vesicles are formed because of the negative curvature of the endosomal membrane, which depends on recognition of ubiquitinated endosomal proteins by ESCRT-0 [14]. Next, ESCRT-I, II and III interact with ESCRT-0 to enable intraluminal vesicle formation in multivesicular endosomes [14]. Lastly, the AAA ATPase Vsp4 complex causes ESCRT-III to separate from the multivesicular endosomal membrane, and the ESCRT complex is recycled [14].

Several studies have highlighted the existence of an ESCRT-independent pathway for exosome biogenesis, based on lipid raft microdomains located in the late endosome membrane [12]. These microdomains contain cone-shaped molecules (e.g., lysobisphosphatidic acid and ceramides) that can induce negative curvature of the endosomal membrane [14,15].

#### 2.2.2. The Biogenesis of Microvesicles

In contrast to exosomes, microvesicles are generated by direct budding from the plasma membrane. During microvesicle biogenesis, the interaction between the cytoskeleton and the plasma membrane weakens and several proteins (such as calpains or lipid translocases) are activated [12]. This cytoskeletal remodeling has been linked to an increase in the intracytosolic calcium concentration, which, in turn, induces phosphatidylserine externalization, bud formation and microvesicle secretion [16].

#### 2.2.3. The Biogenesis of Apoptotic Bodies

Unlike exosomes and microvesicles (which are secreted during normal cellular processes), apoptotic bodies are only released during apoptosis [12]. First, the cell membrane buds. Next, externalization of phosphatidylserine (as observed during microvesicle biogenesis) leads to cytoskeletal remodeling and then the formation of apoptotic bodies [12].

### 2.3. Secretion of EVs

Several different molecules are involved in the secretion of EVs in general and exosomes in particular, since microvesicles and apoptotic bodies are generated directly by the budding of the plasma membrane.

The Ras-related proteins in brain (Rab) family is composed of more than 60 GTPases [17]. Screening studies have identified a number of small GTPases proteins involved in exosome secretion in various cell-based models [17]. Rab GTPases are involved in (i) the movement of multivesicular endosomes from the cytoplasm to the plasma membrane and (ii) the fusion of the cell’s plasma membrane with that of exosome-containing multivesicular endosomes [17].

Other molecules are involved in exosome secretion [17]. Thus, soluble n-ethylmaleimide-sensitive-factor attachment protein receptor complexes are thought to be involved in exosome secretion in various cell types [17]. The interaction between soluble n-ethylmaleimide-sensitive factor attachment protein and its receptor leads to the fusion of the plasma membrane with the multivesicular endosome membrane [17]. Some studies have identified other molecules with a potential role in exosome secretion, including diacyl glycerol kinase α, which appears to downregulate multivesicular endosome formation [18]. Citron kinase (a RhoA effector) and the V0 subunit of V-ATPase appear to be also involved in exosome secretion [19,20].

It is noteworthy that the exosome secretion pathway depends on the EVs’ cellular origin.

### 2.4. Fate of EVs

The release of EVs from parental cells may interact with target cells and influence target cell behavior and phenotype behavior [21]. Indeed, EVs carry bioactive molecules such as proteins, lipids and nucleic acids, which have been shown to impact target cells via several mechanisms: (1) direct stimulation of the target cells upon binding to cell surface; (2) transfer of activated receptors to recipient cells; and (3) epigenetic reprogramming of recipient cells via the delivery of functional protein, lipids and RNA.

After EVs are released, they can interact directly with recipient cells by binding to cell surface integrins, proteoglycans or extracellular matrix components, thus inducing different biological processes [1]. For instance, intracellular adhesion molecule-1 on exosomes can interact with lymphocyte function-associated antigen-1 on dendritic cells [22]. Similarly, milk fat globule-EGF factor 8 protein (a lactadherin precursor present on immune cells) can interact with the phosphatidylserine on exosomes [22]. These interactions allow exosome internalization, the presentation of exosome-derived peptides to T cells and T cell activation [22]. Several studies have identified other molecular interactions between EVs and recipient cells in various cellular models [17].

The EV membrane can also merge with the plasma membrane of recipient cells, thus releasing the EVs’ contents (miRNAs, proteins, peptides, nucleic acids, etc.) into the recipient’s cytoplasm [1]. EVs are particularly enriched with miRNAs loaded into EVs via RNA binding protein recruitment [23]. For instance, synaptotagmin-binding cytoplasmic RNA-interaction protein (SYNCRIP) can be associated with miR-3470a and miR-194-2-3p [23]. Other RNA binding proteins are implicated in the sorting process for miRNAs into EVs, such as argonaute2 protein (Ago2) or Y-box binding protein 1 (YBX-1) [23]. The uptake of EVs by the recipient cell occurs through a variety of processes, such as micropinocytosis, endocytosis and phagocytosis (for a review, see References [1,24]). After release, the EVs’ contents can induce various biological processes [17]. For instance, some EVs are enriched in enzymes such as cyclooxygenase and thromboxane synthase, which can regulate platelet activation and aggregation by metabolizing arachidonic acid into thromboxane [25].

After their release, EVs can also be found in various biological fluids, such as blood or urine. Although the half-life of EVs has not been determined, EVs appear to be stable in biological fluids for at least several hours [17]. Moreover, intravenously injected EVs can be found in many organs (such as the spleen, lung and liver) [17].

## 3. Extracellular Vesicles in a Physiological Setting

### 3.1. General Biological Functions of EVs

EVs are involved in various physiological processes, as summarized in Figure 2. The vesicles not only ensure cellular homeostasis but are also involved in intercellular communication and signal transduction as carriers for lipids, RNA, proteins and micro RNAs (miRs) [26]. Furthermore, as EVs can be found in many biological fluids in humans (such as urine, saliva, bile, breast milk, seminal plasma and synovial, cerebrospinal, nasal, bronchoalveolar, uterine and amniotic fluids), they can potentially interact with many different types of cell [27].

EVs are known to have a major role in vascular physiology in general and blood coagulation and angiogenesis in particular [27,28]. Indeed, it has been shown that EVs are involved in reticulocyte maturation and can act as procoagulant, anticoagulant or pro-fibrinolytic agents, depending on the cellular origin [27]. EVs also modulate innate and adaptive immune responses [27]. The data on the role of EVs in inflammation are subject to debate. Indeed, many studies have found EVs to be pro-inflammatory mediators [27]. EVs can transport a wide variety of cytokines, such as interleukin-1β, tumor necrosis factor α (TNF-α) or interferons [29,30]. Moreover, they are involved in the activity of the nitric oxide dismutase-like receptor family pyrin domain containing 3 inflammasome [31]. In contrast, several studies have highlighted a cell-origin-dependent inhibitory role of EVs in inflammation and immune responses [27]. For instance, the cargos of EVs secreted by neutrophilic granulocytes lead to the inhibition of immune cells like monocytes [27]. EVs have other important biological functions that are well described in several reviews [27,32], such as oxidative stress [33], tissue repair [34] or apoptosis [35].

### 3.2. Biological Functions of EVs, According to Their Cellular Origin

#### 3.2.1. Endothelial-Cell-Derived EVs

Endothelial-cell-derived EVs constitute a large population of EVs. They regulate various physiological functions, including vascular homeostasis in particular [36]. These EVs are secreted during cell activation or apoptosis and are involved in endothelial cell survival [36]. Endothelial-cell-derived microvesicles can interact with many types of vascular cell and can induce pro-coagulative or pro-inflammatory responses [36,37]. Endothelial-cell-derived EVs carry a broad spectrum of biomolecules, such as miRs, which can induce various intracellular pathways involved in angiopoiesis, neovascularization, tissue regeneration, cytoprotection and wound healing [38].

#### 3.2.2. Platelet-Derived EVs

Platelet-derived EVs are the most abundant EVs in human blood [39]. Like endothelial-cell-derived EVs, platelet-derived EVs are involved in coagulation, tissue repair, immune responses, inflammation, angiogenesis and wound healing by virtue of their bioactive contents (e.g., growth factors and cytokines) [39]. Platelet-derived EVs are also involved in waste management [40]. Lastly, these EVs can reduce internal stress by encapsulating active caspase-3, a protein involved in apoptosis [40].

#### 3.2.3. Neutrophil- and Leukocyte-Derived EVs

Neutrophil- and leukocyte-derived EVs have major impacts on immune responses and vascular homeostasis [37,41]. Leukocyte-derived EVs can originate from monocytes, neutrophils, B cells or T cells [37,41]. In contrast, neutrophil-derived EVs originate from neutrophils only [42]. There are two subgroups of neutrophil-derived EVs: neutrophil-derived trails (NDTRs), generated from migrating neutrophils that interact with endothelial cells during blood vessel extravasation, and neutrophil-derived microvesicles (NDMVs), secreted by migrated neutrophils [42]. Although the NDTRs’ functions have not been fully elucidated, some studies have revealed that they can guide immune cells (such as T cells) to inflammatory foci [42]. In contrast, NDMVs have various known functions; they can generate reactive oxygen species and induce bactericidal and pro- or anti-inflammatory responses [42]. Other studies have identified a major role of neutrophil-derived EVs in the endothelial barrier function in general and endothelial permeability in particular [43,44]. Indeed, depending on their contents, neutrophil-derived EVs can have either protective or disruptive effects on the endothelial barrier [43].

#### 3.2.4. VSMC-Derived EVs

VSMC-derived EVs and VSMC-derived exosomes have important physiological roles in vascular repair [45]. These EVs are enriched in various biomolecules (such as miR-143, integrins and extracellular matrix proteins) and therefore have an impact on cell proliferation and migration [45]. Furthermore, an LC-MS/MS proteomic analysis of VSMC-derived EVs found that VSMC-derived microvesicles are enriched in cytoplasmic proteins, organelle-associated proteins and housekeeping factors, whereas VSMC-derived exosomes contain more cell adhesion and extracellular matrix proteins [46]. Thus, depending on the content, VSMC-derived exosomes are more likely to be involved in vascular remodeling by mediating cell–cell signaling, contractile cell interaction and VSMC phenotype switching [46].

#### 3.2.5. Bone-Cell-Derived EVs

Bone-cell-derived EVs have various biological effects on human physiology [47,48]. These EVs are involved in bone remodeling and regeneration by promoting cell–cell communication between osteoclasts, osteoblasts and osteocytes via the transport of biomolecules [47,48]. For instance, the miR-214-3p carried by osteoclast-derived exosomes can inhibit osteoblast activity [48]. Furthermore, Li et al. hypothesized that bone-cell-derived EVs are involved in inter-organ communication [48]. EVs can transport various factors secreted by bone cells and act as paracrine modulators of organ functions [48].

#### 3.2.6. Stem/Progenitor-Cell-Derived EVs

Although stem/progenitor cells have promising therapeutic applications, the use of this cell type also has drawbacks, such as immunogenicity and safety limits. Hence, EVs secreted by stem/progenitor cells have recently attracted much scientific interest. Many studies have described the immunomodulatory, anti-inflammatory and regenerative effects of MSC-derived EVs [49]. Moreover, the bioactive molecules carried by EVs derived from embryonic stem cells are able to induce hematopoietic stem cell self-renewal [50]. Furthermore, the EVs released by endothelial progenitor cells have proangiogenic effects [50]. Several other studies have highlighted additional physiological functions of stem/progenitor-cell-derived EVs (for a review, see Reference [50]).

### 3.3. Preparation of EVs

The International Society of Extracellular Vesicles (ISEV) recently published their recommendations and guidelines proposing minimal information requirements for EVs studies [51]. Several techniques have recently been developed to isolate EVs. EVs can thus be isolated from various fluids (cell culture medium, plasma and urine) using different methods such as ultracentrifugation, density gradient, size-exclusion chromatography, filtration and nano-flow cytometry [52]. All of these techniques have both advantages and disadvantages, but ultracentrifugation remains the gold-standard technique for EVs isolation [52]. After isolation, the characterization of EVs can be assessed using either physicochemical methods based on microscopy and imaging, such as nanoparticle tracking analysis (NTA), or biochemical and molecular methods by analyzing the expression of EVs markers [52]. Thus, the ISEV recommends, for general characterization of EVs, the use of three specific EVs markers, including at least one transmembrane/lipid-bound protein such as non-tissue specific tetraspanins (CD63, CD81, CD82) and tissue specific acetylcholinesterase (neurons), for example; and one cytosolic protein recovered in EVs (ESCRTI/I/III, flotillins, caveolins, etc.). This characterization should also include at least one non-EVs protein marker, such as lipoproteins, albumin, etc., that can be recovered in the EVs preparation.

## 4. Extracellular Vesicles in a Pathological Setting: A Focus on CKD and VC

### 4.1. EVs and CKD

Many studies have found that EVs are key players in several human diseases. Indeed, as research into EVs has grown, databases like EVpedia, Vesiclepedia and ExoCarta have been created as repositories for the latest information on EVs in prokaryotes and eukaryotes [53,54,55]. Although EVs appear to be associated with many diseases, their involvement in renal diseases (e.g., CKD) is especially strong (Table 1 and Table 2). Several studies have also highlighted a major role for miRNA transported by EVs in CKD, as summarized by Abbasian et al. [56]. Recently, Zietzer et al. have identified an alteration of intercellular communication mediated by miRNA through EVs in CKD patients, thus promoting endothelial dysfunction [57]. Furthermore, depending on their cell origin, EVs are linked to various pathophysiological processes in CKD.

#### 4.1.1. Endothelial-Cell-Derived EVs

Firstly, many clinical studies have shown that CKD patients have abnormally high levels of EVs in general and endothelial microvesicles in particular. Indeed, endothelial microvesicles are found to be higher in many cohorts of patients with end-stage renal disease (ESRD) [58,59,60,61,62,63]. A cohort of 100 hypertensive patients also presented an elevated circulating level of endothelial microvesicles; this was associated with a low glomerular filtration rate (GFR) [64]. Furthermore, it is known that plasma endothelial microvesicles are strong predictors of cardiovascular diseases. As such, they can be used as markers of endothelial dysfunction, atherosclerosis and arterial stiffness in CKD patients [58]. Elevated levels of apoptotic endothelial microvesicles were also found in patients with ESRD, especially in haemodialyzed CKD patients [60]. Moreover, the level of endothelial microvesicles in haemodialyzed patients is inversely correlated with laminar shear stress, which is a major determinant of plasma endothelial microvesicle levels in ESRD [65].

#### 4.1.2. Platelet–Derived EVs

Other studies found that levels of platelet-derived microvesicles were also elevated in CKD patients [66] and that these EVs appear to have procoagulant and prothrombotic activity [67,68]. In vivo, a higher blood level of platelet-derived microvesicles was also observed in a mouse model of CKD [69]. Furthermore, circulating levels of microvesicles were higher in CKD stage 5 patients with diabetes mellitus than in those without diabetes mellitus [70]. Circulating levels of microvesicles were also significantly elevated in CKD patients with coronary artery disease [71]. Furthermore, Benito-Martin et al. reported on elevated urine levels of exosomal proteins in CKD patients [72]. The podocyte marker CD2AP was found to be downregulated in urinary exosomes from CKD patients (relative to healthy controls), and its levels were linked to the severity of renal fibrosis [73].

#### 4.1.3. Neutrophil–Derived EVs

In a study of CKD patients, Daniel et al. did not find an increase in neutrophil-derived microvesicles or a correlation between microvesicle release and the creatinine clearance rate [74]. Patients with ESRD reportedly have low levels of exosomes, but this was determined solely by CD63 immunoblotting and must be confirmed by quantifying other exosome markers (such as CD9 and CD81) [75].

**Table 1 toxins-12-00811-t001:** Clinical studies on extracellular vesicles in chronic kidney disease.

	Subtype of EVs	EVs Origin	Study Population		Studied Parameters	EVs Effects	References
			Stage of CKD (n)	Therapeutic			
**Clinical studies**	**MV**	Endothelial cells	-	-	-	↑ MV levels	[58,59,60,61,62,63,64,66,68,70,71,76]
			I-II-III (100)	Anti-hypertensive therapy	MV levels by FACS Renal function (estimated GFR, hsCRP, NT-proBNP)	↑ MV levels associated with a decrease of GFR	[64]
			ESRD (227)	Kidney transplantation	MV levels by FACS	↓ MV levels after kidney transplantation	[77]
			ESRD (52)	Kidney transplantation HD before graft	MV levels by flow cytometry	↓ MV levels after kidney transplantation	[78]
			ESRD (81)	HD	Global and cardiovascular mortality (fatal myocardial infarction, stroke, acute pulmonary oedema and sudden cardiac death)	High predictors of cardiovascular outcome	[59]
			ESRD (37)	Dialysis (HD, PD)	cIMT and PWV by high-resolution ultrasound GFR, blood pressure, fasting lipid profile CRP, PTH, BUN, hemoglobin, albumin, serum creatinine, calcium and phosphorus levels	Markers of atherosclerosis and arterial stiffness	[58]
			ESRD (33)	Pre-dialysis			
			ESRD (34)	HD	Arterial hemodynamic measurements (blood pressure and viscosity, brachial artery and aortic shear stress, hematocrit)	MV levels inversely correlated with laminar shear stress	[65]
			ESRD (44)	HD	Arterial function analysis (FMD, CCA intima-media thickness, pulse pressure, distensibility and augmentation index, CCA and brachial artery diameters and pressures, wall motion, aortic PWV)	MV levels associated with endothelial and arterial dysfunction	[61]
			(30) III-IV (30)	-	Brachial artery FMD Complement fragment and alternative pathway activity	Activation of the alternative complement pathway in vitro	[79]
			IV (8)	-	Thrombin generation by CAT Plasma markers of endothelial activation quantification by ELISA	Less procoagulant	[63]
			(10) ESRD (9)	HD			
				PD			
		Platelets	(20) ESRD (17)	HD	Thrombin generation by CAT	Prothrombotic and procoagulant	[68]
				PD			
		Neutrophils	(135)	-	MV levels by flow cytometry Creatinine clearance	No correlation between MV release and creatinine clearance	[74]
			ESRD (40)	HD			
	**Exosomes**	Neutrophils	III (15)	-	CD63 quantification	↓ Exosomes levels	[75]
			IV (18)				
			ESRD (20)	HD			
		CKD patient urine	(32)	-	CD2AP exosomes gene expression Renal function (estimated GFR, BUN, proteinuria, serum creatinine, tubulointerstitial fibrosis and glomerulosclerosis)	↓ podocyte marker CD2AP	[73]
			(14)	-	OPG protein expression Exosomes proteomic analysis by LC-MS/MS OPG identification in exosomes by SRM	↑ inflammatory marker OPG	[72]

BUN: blood urea nitrogen, CAT: calibrated automated thrombography, CCA: common carotid artery, CD2AP: CD2-associated protein, cIMT: carotid artery intima-media thickness, CKD: chronic kidney disease, CRP: C-reactive protein, ELISA: enzyme-linked immunosorbent assay, ESRD: end stage renal disease, EVs: extracellular vesicles, FACS: fluorescence activated cell sorting, FMD: flow–mediated dilation, GFR: glomerular filtration rate, HD: hemodialysis, hsCRP: high-sensitivity C-reactive protein, MV: microvesicles, NT-proBNP: N terminal-pro-brain natriuretic peptide, n: number, OPG: osteoprotegerin, PD: peritoneal dialysis, PTH: parathyroid hormone, PWV: pulse wave velocity, SRM: selected reaction monitoring. ↓: decrease and ↑: increase.

**Table 2 toxins-12-00811-t002:** Preclinical studies on extracellular vesicles in chronic kidney disease.

	Subtype of EVs	EVs Origin	CKD Model	Animal (n)	Studied Parameters	EVs Effects	References
**Preclinical studies**	**MV**	Blood	5/6 nephrectomy	Mice (6)	MV levels by FACS	↑ MV levels	[69]
	**Exosomes**	Urine		Rats (16)	Exosomes quantification by NTA and Western blotting (Alix, CD63, CD9) CCL2 expression in kidney exosomes by RT-PCR	↑ levels of exosomes containing CCL2 promoting inflammatory kidney injury	[80]

CCL2: chemokine ligand 2, CKD: chronic kidney disease, EVs: extracellular vesicles, FACS: fluorescence activated cell sorting, MV: microvesicles, NTA: nanoparticle tracking analysis, n: number RT-PCR: real time-polymerase chain reaction, ↑: increase.

### 4.2. Effect of Haemodialysis on EVs

CKD progression leads to ESRD, which requires renal replacement therapy. Several dialysis techniques are currently used, including haemodialysis (HD), haemofiltration (HF), haemodiafiltration, and peritoneal dialysis (PD). HD is the most common renal replacement therapy for CKD patients. It effectively removes free small molecular weight compounds but not protein-bound molecules or large molecules. The results of the few studies of the effects of HD on EVs are subject to debate. Indeed, some studies found that plasma EV levels are higher in haemodialyzed patients than in healthy subjects [60,62,74,81]; this might be due to an increase in haemodynamic stress during HD. It is noteworthy that the types of altered microvesicle vary from one study to another. Thus, Daniel et al. observed an increase in the level of neutrophil-derived microvesicles but not of platelet-derived microvesicles [74], while Faure et al. reported an increase in the level of platelet microvesicles but not of endothelial or leukocyte microvesicles [62]. In contrast, some studies observed a fall in the level of EVs during HD [82,83]. For example, Ruzicka et al. showed that levels of all circulating microvesicle populations (i.e., endothelial-, leukocyte- and platelet-derived EVs and total EVs) were reduced by HD [83]. This was in line with Georgatzakou et al.’s observation of low post-dialysis levels of total and red-blood-cell-derived microvesicles [82]. By analyzing protein markers, Ruzicka et al. showed that the observed reduction in circulating EVs was due to adsorption of the EVs to the dialysis membrane rather than ultrafiltration [83]. In online HF, molecules are removed from the blood by ultrafiltration of a large blood volume under high hydrostatic pressure through a larger pore size membrane. Online HF has been shown to decrease cardiomortality [84] and inflammation [85] in ESRD patients. Moreover, Cavallari et al. showed that online haemodiafiltration is better than HD because it decreased inflammation and miR223 expression in EVs, thus reducing VSMCs calcification and improving angiogenesis by human umbilical vein endothelial cells (HUVECs) [86]. In conclusion, the choice of the method for renal replacement therapy is critical for EVs.

### 4.3. Impact of Uremic Toxins on EVs

In addition to traditional cardiovascular risk factors, CKD patients also have non-traditional CKD-specific cardiovascular risk factors, such as the accumulation of uremic toxins. Indeed, as kidney function decreases, uremic toxins accumulate in the body fluids of CKD patients. Many studies have reported abnormally high concentrations of uremic toxins in the serum of CKD patients [87]. The European Uremic Toxin Work Group (EUTox) provides researchers and clinicians with relevant information on uremic toxins, especially in relation to CKD [88]. More than 130 solutes are listed in the EUTox database (www.uremic-toxins.org), and most of these are associated with the occurrence of cardiovascular and nephrological disorders.

Uremic toxins can be classified into three groups as a function of their biochemical and physical properties. The first class is composed of water-soluble, low-molecular-weight compounds, such as urea. The second group contains middle-sized compounds, such as 2-microglobulin. Lastly, the third class comprises protein-bound molecules, such as indoxyl sulphate (IS) and p-cresyl sulphate (pCS). The majority of uremic toxins are water-soluble solutes, but the protein-bound toxins are the most difficult solutes to remove by dialysis in CKD patients because of their strong binding to proteins. The most intensively studied uremic toxins in CKD are inorganic phosphate, IS, pCS and homocysteine.

EVs appear to be linked to the physiopathology of CKD. Indeed, several studies have highlighted a link between the accumulation of uremic toxins in CKD patients and EV release [89]. Usually, EV secretion is elevated under uremic conditions. Thus, the level of platelet-derived microvesicles was higher in a cohort of 18 uremic patients than in a group of healthy subjects [90]. Gao et al. also reported that uremic patients had an elevated number of circulating microvesicles derived from peripheral blood cells and endothelial cells [91]. Furthermore, various animal- and cell-based studies have found that IS, pCS and homocysteine enhance the release of EVs in vitro and in vivo [92,93,94,95,96,97,98,99,100]. IS can increase the release of EVs by HUVECs, which leads to the induction of inflammation, apoptosis, cellular senescence, proliferation, calcification and neointimal hyperplasia [98,100,101,102]. Furthermore, IS can promote thrombosis through the release of procoagulant platelet microvesicles [103]. Inorganic phosphate induces the release of procoagulant microvesicles by human endothelial cells [104]. After treatment with 2.5 mM inorganic phosphate, human coronary artery endothelial cells released more annexin II-positive microvesicles than control cells did [105]. Shang et al. showed that uremic toxins like IS accumulated in CKD patients and could induce oxidative stress and thus an increase in the release of microvesicles containing miR-92a (a miR responsible for endothelial dysfunction) [106].

### 4.4. Role of EVs in VC

Several studies have demonstrated that soluble factors secreted by endothelial cells regulate pro-calcifying activity in VSMCs [107,108]. Mature endothelial cells are capable of producing EVs in response to cellular activation, inflammatory stimuli or apoptosis [109], a situation that has also been observed in CKD [62,110]. Under inflammatory conditions, endothelial cells can package into EVs many inflammation-related miRNAs or inflammation-induced molecules that can target perivascular cells to increase the expression of platelet-derived growth factor (PDGF) [111], which is involved in the development of VC through oxidative stress, inflammation and osteogenic switching [112]. BMP2 has been also shown to be one of the encapsulated proteins [113]. In addition to inflammatory stimuli, senescent endothelial cells were shown to release large amounts of EVs that contained calcium ions and BMP2, which serve as nucleation sites for calcification [114].

EVs are known to mediate the VC observed in CKD patients (Figure 3) (Table 3) [115,116]. The EVs’ mode of action during VC has not been fully elucidated because the biological effects depend on the type of EV and the EVs’ cellular origin [117].

#### 4.4.1. VSMC-Derived EVs

In many studies, VSMC-derived EVs appear to be significantly involved in the physiopathology VC [133]. Calcifying EVs released from VSMCs are involved in VSMC osteogenic differentiation and in collagen matrix calcification via dysregulation of the phosphate-calcium balance [45,134,135]. VSMC-derived exosomes are able to trigger the formation of a complex between externalized phosphatidylserine and annexin A6, which induces VC [45]. Furthermore, EVs can interact with collagen via various proteins (such as integrins α1β1 and α2β1) by binding to the GFOGER amino acid sequence found in type I collagen [136,137,138]. Moreover, an oligogalacturonic acid (DP8) was able to block the GFOGER interaction between type I collagen and EVs and thus to reduce VC and VSMC osteogenic switching [128]. These findings were confirmed by Mansour et al., who showed that a synthetic GFOGER peptide could inhibit VC in VSMCs and aortic rings [129]. This effect was accompanied by a change in the EVs’ protein content [129].

#### 4.4.2. Valvular-Interstitial Cell-Derived EVs

Some studies have highlighted the role of EVs derived from valvular interstitial cells (VICs) in VC [121,125,130]. Thus, Cui et al. performed a proteomic analysis of EVs from calcified VICs and identified the up-regulation of calcification regulators, such as calcium-binding annexins [130]. The researchers also highlighted the co-localization of annexin VI with calcifying VIC-derived EVs [130]. Furthermore, Bouchareb et al. found that mineralization of the aortic valve after mechanical strain was promoted by spheroid microvesicles derived from VICs [125]. Lastly, Rogers et al. reported that annexin A1 tethering of VIC-derived EVs induced microcalcification and EV aggregation [121].

#### 4.4.3. Endothelial-Cell-Derived EVs

Endothelial-cell-derived EVs also appear to be involved in the pathogenesis of VC. CKD patients with VC have more circulating endothelial microvesicles than those without VC [118]. These microvesicles are able to increase osteocalcin expression in endothelial progenitor cells, VSMCs and fibroblast cells from healthy donors and especially from CKD patients with VC [118]. Buendía et al. showed that endothelial microvesicles harvested from CKD patients stimulated calcification and osteogenesis by VSMCs [123]. Furthermore, Alique et al. have shown that microvesicles produced by senescent endothelial cells can trigger the development of VC in elderly adults [124]. EVs from a cohort of CKD patients also induce VC of VSMCs, whereas Gla-rich protein (known to have anti-inflammatory activity) appears to inhibit this effect [120]. Furthermore, the plasma level of endothelial microvesicles in CKD patients decreased after kidney transplantation [77,78].

#### 4.4.4. Macrophage–Derived EVs

Macrophages have an important role in VC [139]. Studies have identified macrophage-derived EVs as mediators of VC [131,132,140]. New et al. have shown that macrophage-derived EVs can promote microcalcification in CKD [131]. These EVs are enriched in annexin V and in S100A9, a calcium-binding protein significantly involved in inflammatory processes and that mediates mineralization [131]. The researchers hypothesized that annexin V and S100A9 form a complex with phosphatidylserine in the EVs membrane, which, in turn, would become a nucleation site for hydroxyapatite and then microcalcification [131]. Furthermore, Chen et al. highlighted a role for macrophage-derived EVs in ectopic mineralization [132]. In fact, the cytokine high mobility group box 1 can increase the secretion of EVs from macrophages via the RAGE/p38 MAPK/nSMase2 signaling pathway [132]. These macrophage-derived EVs were then able to promote mineralization in vitro and in vivo [132]. Furthermore, Nguyen et al. reported that macrophage-derived EVs can promote the development of atherosclerosis, a disease that is tightly linked to VC [140]. Indeed, EVs from atherogenic macrophages can transfer miR-146a to naïve macrophages, inducing a decrease in cell migration, an increase in macrophage entrapment and thus atherosclerosis [140].

## 5. Extracellular Vesicles as Biomarkers of CKD

Conventional biomarkers for CKD diagnosis include a low GFR and albuminuria [141]. Recent studies have focused on urinary EVs since they may be non-invasive diagnostic or prognostic biomarkers [3].

EVs in urine are of great interest as renal disease markers because they reflect the nephron’s physiopathological status [142,143,144,145,146,147,148,149]. Indeed, Pisitkun et al.’s proteomic analysis showed that most EVs in urine come from glomerular and tubular cells because circulating EVs cannot cross the filtration barrier under physiological conditions [142]. Lv et al. suggested that urinary exosomes containing the inflammatory chemokine CCL2 mRNA are biomarkers of CKD [80]. Indeed, elevated levels of these exosomes were observed in the kidney and urine of CKD rats and patients with IgA nephropathy [80]. Exosomes can transfer CCL2 mRNA from tubular epithelial cells to macrophages, promoting inflammatory kidney injury [80]. Furthermore, Lv et al. showed that mRNA encoding CD2AP in urinary exosomes was correlated with kidney function and renal fibrosis [73].

Clinical and preclinical studies have highlighted the value of miRs (transported by EVs) as CKD biomarkers [56,106,150,151,152,153,154,155,156,157,158,159]. Indeed, exosomes of patients with kidney diseases contain high levels of miR [152]. Liu et al. have shown that urinary miR-126 (probably secreted by exosomes) is more strongly expressed in patients with diabetic nephropathy, whereas treated patients have lower expression levels of this miR [154]. A specific urinary exosomal miR profile was found in patients with focal segmental glomerulosclerosis, a disease that can lead to CKD [156]. Furthermore, miR29 and miR200 are downregulated in urinary exosomes from CKD patients [159]. The miR-451-5p and miR-16 found in urinary exosomes from diabetic rats also appear to protect the kidney tissue [158].

Some researchers provide technical information about urinary EVs and miR transported by these EVs as non-invasive biomarkers for kidney diseases such as CKD [152,153]. However, it is important to note that EVs cannot always be handled easily and that the creation of standardized protocols is required if EVs are to be used as biomarkers in the clinic.

## 6. The Therapeutic Potential of Extracellular Vesicles in CKD

As described above, EVs are involved in the physiopathology of CKD, VC and uraemic conditions. Therefore, EVs are potential therapeutic targets that could be inhibited by various pharmacological agents [2,160]. EVs could also be used as drug delivery systems because of their involvement in cell–cell communication [161].

### 6.1. Inhibition of EVs

Several in vitro studies have demonstrated that various molecules can be used to block EV release or the uptake of EVs by recipient cells, as reviewed elsewhere [2,160].

The complexity and heterogeneity of EV biogenesis complicates the development of drugs that inhibit EVs. In principle, inhibitors could act at several different stages of EV biogenesis. The Rab family is involved in exosome secretion [17]. It has been shown that the inhibition of Rab proteins (Rab27A and Rab27B) leads to a decrease in exosome secretion [162]. Several other inhibitors (including calpeptin, Y27632 and manumycin A) affect EV trafficking [160]. The best studied compound is calpeptin, a reversible, semi-synthetic peptidomimetic aldehyde inhibitor of calpains [160]. Calpains is a family of calcium-dependent cytosolic proteases the action of which on cytoskeletal remodeling promotes microvesicle shedding [12]. Therefore, the inhibition of calpains leads to a decrease in microvesicle release by cells [163].

EV release can also be inhibited by blocking lipid metabolism [160]. As described above, lipid metabolism in general and ceramide metabolism in particular are involved in EV biogenesis [17]. One such inhibitor is panthetine. It inhibits cholesterol synthesis by changing the membrane fluidity and thus blocking the translocation of phosphatidylserine to the outer leaflet, an essential process in microvesicle production [160]. Sphingomyelinases are important enzymes for exosome and microvesicle formation and therefore constitute potential therapeutic targets [160]. Thus, treatment with imipramine or GW4869 (inhibitors of acid sphingomyelinase and membrane neutral sphingomyelinase, respectively) leads to a reduction in EV secretion [160]. Other inhibitors (such as antiplatelet molecules, antioxidants, statins, calcium-channel blockers and proton pump inhibitors) also reduce EV biogenesis (for detailed reviews, see References [2,160]).

Another way of counteracting the EVs’ effects is to reduce their uptake by recipient cells [2]. Even though the inhibition of EV uptake increases the levels of circulating EVs, it will still reduce the harmful effects in recipient cells. This inhibition can be either achieved with antibodies targeting various molecules (e.g., Annexin V and integrin αvβ3) or with drugs that modify, for example, microfilament formation [2].

These inhibitors must be used with caution, however. Special attention should be paid to their mechanism of action and side effects to ensure that these drugs do not reduce EV biogenesis in or uptake by healthy cells.

### 6.2. The Therapeutic Potential of MSC- and EPC-Derived EVs

Given that MSCs have beneficial effects in human diseases like CKD [164], MSC-derived EVs have attracted great interest for their therapeutic potential in CKD. Thus, a protective role for MSC EVs was found in patients with CKD stage 3–4, as evidenced by a higher estimated GFR and lower values for blood urea, serum creatinine and the urine albumin to creatinine ratio [165]. These effects were accompanied by an increase in the plasma levels of anti-inflammatory cytokines (such as TGF-β1 and interleukin-10) and a decrease in the plasma level of TNF-α [165]. Furthermore, MSC-derived microvesicles have been shown to protect rats against CKD and inhibit renal fibrosis [166]. Another in vivo study in mice with CKD induced by subtotal (5/6) nephrectomy showed that MSC-derived EVs were able to decrease not only proteinuria, serum creatinine and uric acid but also fibrosis, interstitial lymphocyte infiltration and tubular atrophy [167]. Kidney MSC microvesicles exert antifibrotic effects by decreasing the endothelial-to-mesenchymal transition and increasing TGF-β-induced HUVEC proliferation [168]. In vivo, the researchers obtained the same results in unilateral ureteral obstruction (UUO) mice and evidenced the inhibition of inflammatory cell infiltration and tubulointerstitial fibrosis [168]. The MSC microvesicles’ protective role was also observed in vitro for proximal tubular epithelial cells, with enhanced E-cadherin expression and lower α-smooth muscle actin (α-SMA) secretion; these results were confirmed in vivo in UUO mice [169]. Furthermore, MSC-derived exosomes containing miR-let7c induce antifibrotic effects, especially with a decrease in expression levels of collagen IVα1, α-SMA and TGF-β1 receptor in vitro [170]. MSC-derived EVs can also reduce renal inflammation and fibrosis and increase medullary oxygenation in pigs with metabolic syndrome and renal artery stenosis [171]. Interestingly, van Koppen et al. have shown that MSC-derived exosomes had no effect on CKD progression, systolic blood pressure and renal damage in 5/6-nephrectomized rats but did stimulate angiogenesis [172]. However, the conditioned, exosome-containing medium in the latter study had protective effects on induced CKD, hypertension and glomerular injury, suggesting that, along with exosomes, other molecules in the conditioned medium were needed to induce protective effects [172].

Circulating endothelial progenitor cells (EPC) play an important role in the maintenance of vascular integrity and the regeneration of the vascular system. The number of circulating EPC in CKD patients is 30% lower than in healthy controls [173,174,175]. Surdacki et al. showed that EPC number is in fact inversely correlated to the degree of kidney dysfunction [176]. The low EPC count [177] and dysfunction [178] may contribute to an increased risk of CVD in CKD patients [179,180]. EPC therapy has been shown to decrease inflammation and proteinuria and to preserve kidney function [181]. This beneficial effect is thought to be mediated in part in a paracrine manner, and recent studies have shown that EPC-EVs may be responsible for this effect by transferring miRNA. Thus, EPC-EVs activate angiogenesis by the transfer of mRNA, especially miR126 and miR296, which have been shown to modulate proliferation, angiogenesis, apoptosis and inflammation [182,183,184]. EPC-EVs also protect against CKD progression after ischemia-reperfusion injury I by inhibiting, in particular, tubule interstitial fibrosis [184].

### 6.3. EVs as Drug Carriers

Thanks to their small size and low immunogenicity, EVs are able to transfer endogenous and exogenous drug compounds (such as siRNA [185], protein drugs or pharmaceutical drugs) into recipient cells. Moreover, the EVs’ lipid bilayer of membrane protects their contents from degradation. The advantages of EVs include their non-toxicity, their long half-life and, importantly, their ability to cross barriers such as the blood-brain barrier [186]. Specific binding of EVs to the target cells is important for effectiveness. To this end, donor cells can be genetically engineered so that a specific ligand is expressed on the EVs they release [187] and can be specifically recognized by the target recipient cells. Most studies have been performed in cell-based or animal models. Only a few drug-carrying EVs have been tested in the clinic [161].

Despite the EVs’ potential as delivery systems, some important issues need to be addressed for clinical use; these include the culture conditions, EV purification/quantification and the choice of the donor cells.

## 7. Conclusions

Here, we reviewed the latest research on the role of EVs in CKD and VC. Levels of EVs are elevated in CKD patients, and several studies have shown that EVs have an important role in VC by inducing VSMC osteogenic switching, inflammation and oxidative stress. Furthermore, many preclinical and clinical studies have shown an association between EV release and the accumulation of uremic toxins (such as IS, pCS and inorganic phosphate) in CKD patients. Our review also highlighted the use of EVs as disease biomarkers (e.g., the miRs contained in urinary EVs) and the potential of MSC-derived EVs for the treatment of CKD patients.

## Figures and Tables

**Figure 1 toxins-12-00811-f001:**
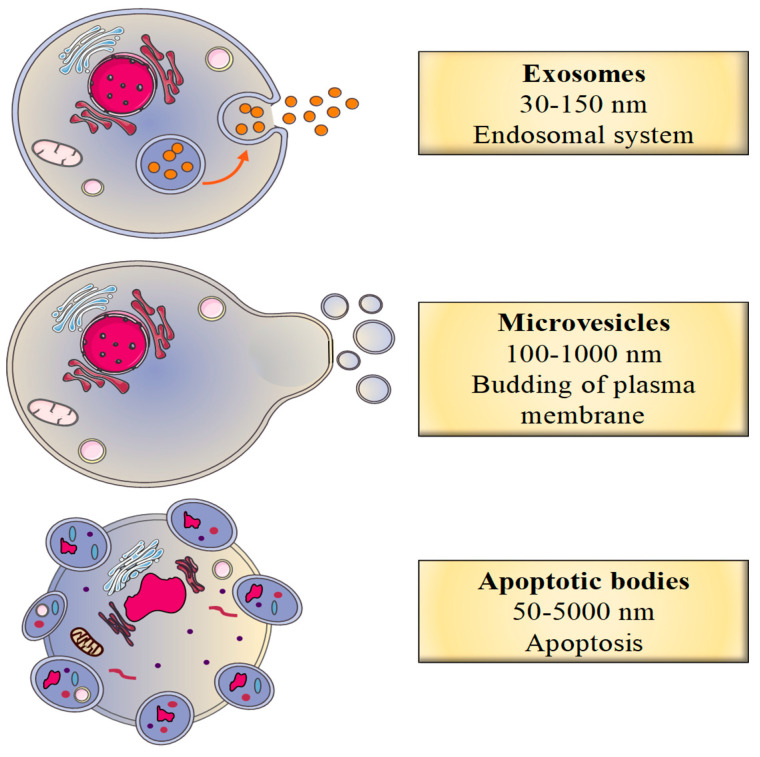
The classification of extracellular vesicles (EVs) as a function of their biogenesis.

**Figure 2 toxins-12-00811-f002:**
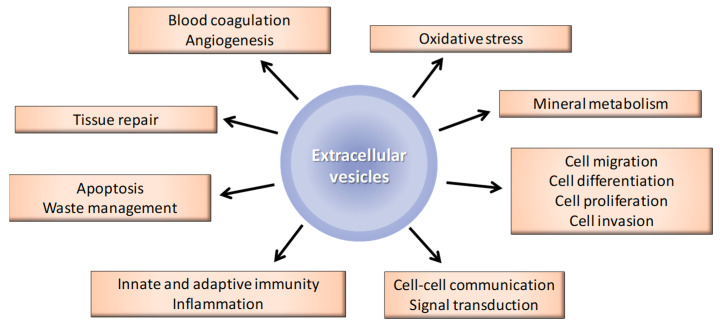
Biological functions of EVs.

**Figure 3 toxins-12-00811-f003:**
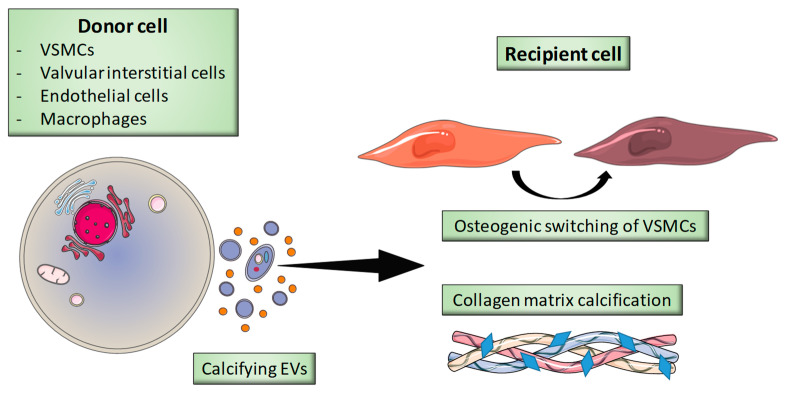
The role of EVs in vascular calcification. VSMCs: vascular smooth muscle cells.

**Table 3 toxins-12-00811-t003:** Extracellular vesicles in vascular calcification.

	Subtype of EVs	Cell Types		Experimental Conditions	Studied Parameters	EVs Effects	References
		EVs Donor Cells	EVs Recipient Cells				
**Clinical studies**	**MV**	Endothelial cells		-	MV levels by FITC-annexin V labelling	↑ MV levels in VC CKD patients	[118]
**Preclinical studies**	**Total EVs**	Human VSMCs		Osteogenic medium 21 days	Visualization of microcalcification by using a bisphosphonate-conjugated imaging agent	Sortilin in EVs mediates VC	[119]
		ESRD patients	Human VSMCs	Phosphate (2.5 mM) 14 days Calcium (5.4 mM) 24 h	Calcium content	Induction of VC	[120]
		Human VSMCs Human VICs		EVs added to 3D collagen hydrogels	Confocal microscopy Collagen staining	Microcalcification formation	[121]
		Human and porcine VSMCs		Ca^2+^ loaded EVs added to collagen matrix	Calcium content	Promote VC by ROS production via NOX5	[122]
	**MV**	IS-treated endothelial cells	Human VSMCs	50,000 MV/mL 30 days	Calcium deposits staining Inflammatory (TNF-α, TWEAK, CCL2, CCL5, and IL-6) and pro-calcification (Runx2, BMP2) gene expression	Induction of VC	[100]
		HUVECs CKD patients	Human VSMCs	Phosphate (2.6 mM) 50 µg/mL MV 5 days	Calcium content Intracellular and MV BMP2 quantification BMP2 gene expression	Stimulation of calcification and osteogenesis	[123]
		VC-CKD patients	EPCs, VSMCs, fibroblast cells	500,000 MV/well	OCN expression by flow cytometry	Increase of OCN expression	[118]
		Elderly subjects’ plasma Senescent endothelial cells	Human VSMCs	50,000 MV/mL 6-9 days	Calcium content Alizarin red staining Annexin A6 and BMP2 protein levels	Promote VC	[124]
		Human and mouse VICs		Mechanical strain of the aortic valve	Calcium content	Promote mineralization	[125]
	**Exosomes**	CKD rat VSMCs	Rat VSMCs	ß-glycerophosphate (5 mM) 10 µg exosomes 7 days	Calcium content Pro-calcification (Runx2, BMP2, OCN), NOX1 and SOD2 gene expression levels by RT-PCR	Induction of VC through NOX1, MEK1 and Erk1/2 signaling	[126]
		Human VSMCs		Calcium (5.4mM)	Flow cytometry, TEM and mass spectrometry analysis of exosomes	Induction of mineralization by formation of a complex between PS on exosomes and Annexin A6	[127]
		Mouse VSMCs		Inorganic phosphate (4 mM) 8 or 14 days	Calcium content Calcification inhibitors and dedifferentiation markers gene expression levels (MGP, OPN, OCN, Runx2, BMP2, TNAP, COL1A1, COL1A2)	Reduction of VC by inhibition of collagen-EVs interaction via GFOGER peptide	[128,129]
		Rat VICs		Calcium-phosphate 5 days	Proteomic and TEM analysis	Up-regulation of calcification regulators (calcium-binding annexins) Co-localization of Annexin VI with exosomes	[130]
		Mouse macrophages		Calcium (3 mM)-inorganic phosphate (2 mM)	Calcium content Alkaline phosphatase activity Immunohistochemical analysis of α-SMA and annexin 5	Promote microcalcification	[131]
		Mouse macrophages		Calcium (1.2 mM)-inorganic phosphate (0.9 mM)	Alizarin red and Von Kossa staining TNAP activity TEM analysis of exosomes OPN, OCN, Runx2, BMP2 and TNAP gene expression	Enhance ectopic mineralization	[132]

α-SMA: α-smooth muscle actin, BMP2: bone morphogenetic protein, CCL2: chemokine ligand 2, CCL5: chemokine ligand 5, CKD: chronic kidney disease, COL1A1: collagen type 1 α 1 chain, COL1A2: collagen type 1 α 2 chain, EPCs: endothelial progenitor cells, Erk1/2: extracellular-signal-regulated kinase 1/2, ESRD: end stage renal disease, EVs: extracellular vesicles, FITC: fluorescein isothiocyanate, HUVECs: human umbilical vein endothelial cells, IL-6: interleukin-6, MEK1: mitogen-activated protein kinase, MGP: matrix Gla protein, MV: microvesicles, NOX1: NADPH oxidase 1, NOX5: NADPH oxidase 5, OCN: osteocalcin, OPN: osteopontin, ROS: reactive oxygen species, RT-PCR: real time-polymerase chain reaction, Runx2: runt-related transcription factor 2, SOD2: superoxide dismutase 2, TEM: transmission electron microscopic, TNAP: tissue nonspecific alkaline phosphatase, TNF-α: tumor necrosis factor-α, TWEAK: tumor necrosis factor-like weak inducer of apoptosis, VC: vascular calcification, VICs: valvular interstitial cells, VSMCs: vascular smooth muscle cells. ↑: increase.

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
