# Peer review of "Effects of Chronic Kidney Disease and Uremic Toxins on Extracellular Vesicle Biology"

_toxins, 2020, doi:10.3390/toxins12120811_

Round 1
Reviewer 1 Report
The Review "Effects of CKD and uremic toxins on EVs’ biology" is quite new and interestin topic. It is well written, even if the paragraphs are sometimes too subdivided into subparagraphs, I can understand it because in fact the topic is complex. I suggest to add 2 tables to the review: one table with the list of all the studies on EVs and CKD, and one table with the list of all studies on EVs related to VC.
Minor:
1) the subparagraphs 2.31 and 2.3.2 could be eliminated and the texts combined in the single paragraph 3.2
2)Pag 4, lines 116-117: for ICAM and LFA please provide full version of initials
3) pag 5, line 148: remove the dot after note 26. Lines 153-154: please, list at least other biological functions
4) Pag 7, line 251: add (PD) after Peritoneal Dialysis
5) pag 10, line 374: Kim-1 is not a classical biomarker for CKD, essentially it is a marker of acute kidney injury
6)Page 11, lines 388-389:Liu et al. ...please clarify the concept you want to express in this sentence.
Lines 398-405: I suggest to eliminate subparagraphs 6.1 and 6.2 to combined all of them in one paragraph 6.1
7) Paragraph 6.2: authors reported that MSCs microvescicles produced an increase of TGF-B1 in CKD patients [147], then a decrease of TGF-B1 in CKD mice [152]: please discuss these inconsistencies.
Author Response
Point-by-point response to Reviewer 1
“The Review "Effects of CKD and uremic toxins on EVs’ biology" is quite new and interestin topic. It is well written, even if the paragraphs are sometimes too subdivided into subparagraphs, I can understand it because in fact the topic is complex. I suggest to add 2 tables to the review: one table with the list of all the studies on EVs and CKD, and one table with the list of all studies on EVs related to VC.”
As reviewer 1 suggested, we added 2 tables: one table with the list of the main and more recent studies on EVs and CKD (Table 1) and one table with the list of studies on EVs related to VC (Table 2). These 2 tables now appear and are referenced in the revised manuscript.
Minor:
- the subparagraphs 2.31 and 2.3.2 could be eliminated and the texts combined in the single paragraph 3.2
2)Pag 4, lines 116-117: for ICAM and LFA please provide full version of initials
3) pag 5, line 148: remove the dot after note 26. Lines 153-154: please, list at least other biological functions
4) Pag 7, line 251: add (PD) after Peritoneal Dialysis
5) pag 10, line 374: Kim-1 is not a classical biomarker for CKD, essentially it is a marker of acute kidney injury
6)Page 11, lines 388-389:Liu et al. ...please clarify the concept you want to express in this sentence.
Lines 398-405: I suggest to eliminate subparagraphs 6.1 and 6.2 to combined all of them in one paragraph 6.1
7) Paragraph 6.2: authors reported that MSCs microvescicles produced an increase of TGF-B1 in CKD patients [147], then a decrease of TGF-B1 in CKD mice [152]: please discuss these inconsistencies.
Regarding the minor modifications, we took into account reviewer 1’s remarks.
Comments 1) and 6):
We have now combined the subparagraphs 2.3.1 and 2.3.2 in a single paragraph 2.3 and subparagraphs 6.1 and 6.2 in a new paragraph 6.1 in the revised manuscript.
Comments 2) and 4):
We added abbreviations when needed and defined the ICAM and LFA abbreviations. These modifications now appear in the body of the manuscript.
Comment 3):
The biological functions of EVs appear in Figure 2 but as the reviewer suggested, we now listed some of them in the text.
Comment 5):
We agree with reviewer 1 and we changed the sentence accordingly in the revised manuscript
Comment 6):
The sentence was changed to: “Liu et al. have shown that urinary miR-126 (probably secreted by exosomes) is more strongly expressed patients with diabetic nephropathy, whereas treated patients have lower expression levels of this miR [158]”
Comment 7):
Paragraph 6.2: authors reported that MSCs microvescicles produced an increase of TGF-B1 in CKD patients [147], then a decrease of TGF-B1 in CKD mice [152]: please discuss these inconsistencies
We agree that there is a discrepancy between these 2 studies regarding TGF1b levels. However, it is noteworthy that the settings of the 2 studies are very different: human vs. mice; intravenous vs. intra renal arterial for the way of administration; number of MSC injections (2 vs. 1) and cell origin of the MSC (umbilical cord MSC vs. bone marrow MSC). The last difference might explain these inconsistencies in the results. Indeed, the secretome of the umbilical cord MSC (UMSC) is different from the one of bone marrow MSC (BMMSC). For example, UCMSC have less proangiogenic factors but more antiangiogenic compared to BMMSC (Amable PR Stem Cell Res dev 2015; Kuchroo P Stem Cell Dev 2015). Moreover the expression of cell surface markers (Maglieri A Int J Clin Exp Med 2010), the proliferative capacity and the function in immune regulation also differ between UCMSC and BMMSC. The origin of the MSC may be of great importance in the biological activities of these MSC and in turn one can speculate that it can also affect the biological activities of the EV-derived from these cells.
Reviewer 2 Report
This is a review paper on the biology of extracelullar vesiclesin CKD. There are two major comments and several minor ones.
Major comments:
- The English language style is quite poor and needs significant improvement.
- On many occasions this review refers to and cites findings from other reviews which dimishes its utility and priority.
Minor comments:
Pg1, the title: it is better to write a title without abbreviations, especially those that are not clear to wide public (e.g. "EV")
Pg 1, line 33: is vascular calcification predictor of CKD or vice versa? Please revise
Pg 1, line 40: "medial" layer is not apropriate term, revise to "media" layer
Pg 1; line 45: it is hard to say that uremic syndrome is a strong predictor of CKD progression (although it is true that some uremic toxins have been associated with progression of CKD) the general view is that uremic syndrome manifests itself in advanced stage of CKD - uremic syndrome is a consequence of CKD and not vice versa
Pg4, line 103: SNARE is given without explanation of this abbreviation. Usually , journals demand that each abbreviation is introduced for the first time in text with explanation followed by specific abbreviation in parentheses. In the same paragraph, line 105, this is done with a reversed orther (SNAP abbreviation is introduced followed by expalanation in parentheses). This should be revised for some other examples in the text below as well.
Pg 4, line 112: poor english language style: "It is worthwile to note that it exists several pathways..:"
Pg 4, line 137: "cargo" or "carrier"?
Pg 5. lines 147-149: two sentences inapropriately separated.
Pg 6, line 190-192: poorly written sentence.
Pg 7, line 221-231: whole paragraph 4.1.1. is poorly written, language style is substandard.
Pg 8,line 271: "HUVECs" abbreviation is not explained.
Pg 8, line 274: "uremic toxins are non-traditional risk factors of CKD" - what exactly does this sentence mean? Please revise.
Pg 10, line 338: "harvest" - "harvested".
Pg 11, line 394: "technical information's" - poor grammar
Author Response
Point-by-point response to Reviewer 2
This is a review paper on the biology of extracelullar vesicles in CKD. There are two major comments and several minor ones.
Major comments:
Comment 1):
The English language style is quite poor and needs significant improvement.
Since reviewer 2 asked for a significant improvement of the English language style, we sent our manuscript to an English editing professional and our revised manuscript now took into account these corrections.
Comment 2):
On many occasions this review refers to and cites findings from other reviews which dimishes its utility and priority.
The topic of EV is such a large and complex field that it is difficult to develop all the findings on EVs that’s why we chose to cite other reviews to help the reader who need more detailed information on a specific topic. As reviewer 2 suggested, we now added some original papers.
Minor comments:
- Pg1, the title: it is better to write a title without abbreviations, especially those that are not clear to wide public (e.g. "EV")
Pg4, line 103: SNARE is given without explanation of this abbreviation. Usually, journals demand that each abbreviation is introduced for the first time in text with explanation followed by specific abbreviation in parentheses. In the same paragraph, line 105, this is done with a reversed other (SNAP abbreviation is introduced followed by explanation in parentheses). This should be revised for some other examples in the text below as well.
Pg 8, line 271: "HUVECs" abbreviation is not explained.
As reviewer 2 requested, we removed all abbreviations present in the title which was changed to “Effects of chronic kidney disease and uremic toxins on extracellular vesicles’ biology”
We made also all the needed changes along the manuscript in order to have now each abbreviation with a given explanation in parentheses. All these changes appear in red in the revised manuscript.
- Pg 1, line 33: is vascular calcification predictor of CKD or vice versa? Please revise
As the reviewer suggested we revised the sentence which was changed to: “Vascular calcification (VC) is a strong predictor of outcomes in CKD. Indeed, it is is associated with high morbidity and mortality rates in CKD patients [7].” (Page 1 line 32-33)
- Pg 1, line 40: "medial" layer is not appropriate term, revise to "media" layerPg 7, line 221-231: whole paragraph 4.1.1. is poorly written, language style is substandard.
- Pg 5. lines 147-149: two sentences inapropriately separated.
Pg 10, line 338: "harvest" - "harvested".
Pg 4, line 112: poor english language style: "It is worthwile to note that it exists several pathways..:
Regarding the English language style, we sent our manuscript to an English editing professional and our revised manuscript now took into account these corrections
The sentence was changed to: “It is noteworthy that the exosome secretion pathway depends on the EVs’ cellular origin.” (Page 3 line 108)
As requested also by the reviewer the whole paragraph 4.1.1. was rewritten and appears in the revised manuscript as follows :
“4.1.1. Endothelial-cell-derived EVs
Firstly, many clinical studies have shown that CKD patients have abnormally high levels of EVs in general and endothelial microvesicles in particular. Indeed, endothelial microvesicles are found higher in many cohorts of patients with end-stage renal disease (ESRD) [59–62]. A cohort of 100 hypertensive patients also presented an elevated circulating level of endothelial microvesicles; this was associated with a low glomerular filtration rate (GFR) [63]. Furthermore, it is known that plasma endothelial microvesicles are strong predictors of cardiovascular disease. As such , they can be used as markers of endothelial dysfunction, atherosclerosis and arterial stiffness in CKD patients [59-61]. Elevated levels of apoptotic endothelial microvesicles were also found in patients with ESRD and especially in haemodialyzed CKD patients [62]. Moreover, the level of endothelial microvesicles from haemodialyzed patients is inversely correlated with laminar shear stress, which is a major determinant of plasma endothelial microvesicle levels in ESRD [64].”
- Pg 1; line 45: it is hard to say that uremic syndrome is a strong predictor of CKD progression (although it is true that some uremic toxins have been associated with progression of CKD) the general view is that uremic syndrome manifests itself in advanced stage of CKD - uremic syndrome is a consequence of CKD and not vice versa
The remark of the reviewer is pertinent and we now changed the sentence for: “The uremic syndrome in patients with advanced CKD can be responsible for many different symptoms including anemia, bone-mineral disorders, and hypertension [12].” These changes are now included in the revised manuscript. (Page 1-2 lines 45-47)
- Pg 8, line 274: "uremic toxins are non-traditional risk factors of CKD" - what exactly does this sentence mean? Please revise.
We apologized because the sentence was indeed confusing. We clarified the meaning of the sentence that is now changed to: “In addition to traditional cardiovascular risk factors, CKD patients have also non-traditional CKD-specific cardiovascular risk factors such as the accumulation of uremic toxins.” (Page 11 Lines 310-311)
Reviewer 3 Report
This is a very complete review on the effects of CKD and uremic toxins on Extarcellular Vesicle (EV) biology. The Authors describe the potential role of EV in different aspects of CKD including osteogenic switch of Vascular Smooth Muscle cells, inflammation, oxidative stress and the association of EV levels with known uremic toxins such as PBUT (IS ans PCS). Moreover, the Authors discuss the potential role of EV as disease biomarkers and the regenerative effects of MSC-derived EV.
The review is well organized and written: I have only some minor points that should be addressed:
-I suggest to describe in more details the potential biological effects of EV on target cells: this may be related to direct transfer of proteins, receptors and particuarlyl genetic material such ss mRNA and miRNA. Since EV are particularly enriched for miRNAs, this focused paragraph should be implemented.
-Mechanisms of vascular calcifications: a major focus on the cross-talk between endothelial and vascular smooth muscle cells should be added. The authors should also report the role of endothelial dysfunction in the interaction with VSMC
-Since EV represent the major point of this review, a paragraph dedicated to the methods of EV isoaltion and characetrization in accordance to ISEV guidelines should be considered.
-Stem cell-induced regeneration: the authors focused mainly on MSC-derived EV. However, there are several other stem cell types used in this setting (other types of MSC, renal resident progenitors, etc.). I suggest to add some reports in particular on endotehlial progenitor cells (EPC) for 2 reasons: 1. EPC number is reduced in advanced stages of CKD and seem to contribute to cardiovascualr alterations in CKD patients; 2. EV derived from EPC have been shown to protect the kidney from AKI and progression toward CKD (e.g. Cantaluppi V et al, Kidney Int, 2012)
-
Author Response
Point-by-point response to Reviewer 3
This is a very complete review on the effects of CKD and uremic toxins on Extracellular Vesicle (EV) biology. The Authors describe the potential role of EV in different aspects of CKD including osteogenic switch of Vascular Smooth Muscle cells, inflammation, oxidative stress and the association of EV levels with known uremic toxins such as PBUT (IS and PCS). Moreover, the Authors discuss the potential role of EV as disease biomarkers and the regenerative effects of MSC-derived EV.
The review is well organized and written: I have only some minor points that should be addressed:
Comment 1) :
-I suggest describing in more details the potential biological effects of EV on target cells: this may be related to direct transfer of proteins, receptors and particularly genetic material such ss mRNA and miRNA. Since EV are particularly enriched for miRNAs, this focused paragraph should be implemented.
We agree with Reviewer 3 that the potential biological effects of EV on target cells are an important topic. Some of these biological effects are already described in paragraph “2.4 Fate of EVs.”
As reviewer 3 suggested, we added a paragraph. This now appears in the revised manuscript as follows:
EV release from parental cells may interact with target cells and influence target cell behavior and phenotype behavior [22]. Indeed, EVs carry bioactive molecules such as proteins, lipids and nucleic acid which have been shown to impact target cells via several mechanisms: 1) direct stimulation of target cells upon binding to cell surface 2) transfer of activated receptors to recipient cells 3) epigenetic reprogramming of recipient cells via delivery of functional protein, lipids and RNA.
Thus, after EVs are released, they can interact directly with recipient cells by binding to cell surface integrins, proteoglycans or extracellular matrix components - thus inducing different biological processes [1]. For instance, intracellular adhesion molecule-1 on exosomes can interact with lymphocyte function-associated antigen-1 on dendritic cells [23]. Similarly, milk fat globule-EGF factor 8 protein (a lactadherin precursor present on immune cells) can interact with the phosphatidylserine on exosomes [23]. These interactions allow exosome internalization, the presentation of exosome-derived peptides to T cells, and T cell activation [23]. Several studies have identified other molecular interactions between EVs and recipient cells in various cellular models [18].
The EV membrane can also merge with the plasma membrane of recipient cells, thus releasing the EVs’ contents (miRNAs, proteins, peptides, nucleic acids, etc.) into the recipient’s cytoplasm [1]. EVs are particularly enriched with miRNA loaded into EVs via RNA binding protein recruitment [24]. For instance, synaptotagmin-binding cytoplasmic RNA-interaction protein (SYNCRIP) can be associated with miR-3470a and miR-194-2-3p [24]. Others RNA binding proteins are implicated in the sorting process for miRNA into EVs such as argonaute2 protein (Ago2) or Y-box binding protein 1 (YBX-1) [24]. The uptake of EVs by the recipient cell through a variety of processes such as micropinocytosis, endocytosis and phagocytosis (for a review, see [1,25]). After release, the EVs’ contents can induce various biological processes [18]. For instance, some EVs are enriched in enzymes such as cyclooxygenase and thromboxane synthase, which can regulate platelet activation and aggregation by metabolizing arachidonic acid into thromboxane [26].
After their release, EVs can also be found in various biological fluids, such as blood or urine. Although the EVs’ half-life has not been determined, EVs appear to be stable in biological fluids for at least several hours [18]. Moreover, intravenously injected EVs can be found in many organs (such as the spleen, lung, and liver) [18].
Comment 2) :
Mechanisms of vascular calcifications: a major focus on the cross-talk between endothelial and vascular smooth muscle cells should be added. The authors should also report the role of endothelial dysfunction in the interaction with VSMC.
We agree with the reviewer that the integrity of the endothelium plays a key role in the pathogenic of vascular disease, including the development of VC. Therefore, we have added a short paragraph in the manuscript:
“Several studies have demonstrated that soluble factors secreted by endothelial cells regulate pro-calcifying activity in VSMCs [110-111]. Mature endothelial cells are capable of producing EVs in response to cellular activation, inflammatory stimuli or apoptosis [112], a situation that has also been observed in CKD [76,113]. Under inflammatory conditions, endothelial cells can package into EVs many inflammation-related miRNAs or inflammation-induced molecules that can target perivascular cells to increase the expression of platelet-derived growth factor (PDGF) [114], which is involved in the development of VC through oxidative stress, inflammation and osteogenic switching [115]. BMP2 has been also shown to be one of the encapsulated proteins [116]. In addition to inflammatory stimuli, senescent endothelial cells were shown to release large amounts of EVs, which contained calcium ions and BMP2 which serve as nucleation sites for calcification [117].”
Comment 3) :
Since EV represent the major point of this review, a paragraph dedicated to the methods of EV isolation and characterization in accordance to ISEV guidelines should be considered.
As Reviewer 3 suggested, we added a paragraph regarding the methods of isolation and characterization of EVs according to ISEV guidelines. This paragraph appears now in the revised manuscript as follows:
“3.3. Preparation of EVs
The International Society of Extracellular Vesicles (ISEV) recently published their recommendations and guidelines proposing minimal information requirements for EVs studies [52]. Several techniques have recently been developed to isolate EVs. EVs can thus be isolated from various fluids (cell culture medium, plasma, urine) using different methods such as ultracentrifugation, density gradient, size-exclusion chromatography, filtration, Nano-flow cytometry [53]. All of these techniques have both advantages and disadvantages, but ultracentrifugation remains the gold-standard technique for EVs isolation [53]. After isolation, the characterization of EVs can be assessed using either physicochemical methods based on microscopy and imaging such as nanoparticle tracking analysis (NTA) or biochemical and molecular methods by analysing the expression of EVs markers [53]. Thus, the ISEV recommends for general characterization of EV the use of three specific EV markers included at least one transmembrane/lipid-bound protein such as non-tissue specific tetraspanins (CD63, CD81, CD82) and tissue specific acetycholinesterase (neurons) for example and one cytosolic protein recovered in EVs (ESCRTI/I/III, flotillins, caveolins….). This characterization should also include at least one non-EV protein marker such as lipoproteins, albumin etc... that can be recovered in the EV preparation.”
Comment 4) :
Stem cell-induced regeneration: the authors focused mainly on MSC-derived EV. However, there are several other stem cell types used in this setting (other types of MSC, renal resident progenitors, etc.). I suggest to add some reports in particular on endothelial progenitor cells (EPC) for 2 reasons: 1. EPC number is reduced in advanced stages of CKD and seem to contribute to cardiovascular alterations in CKD patients; 2. EV derived from EPC have been shown to protect the kidney from AKI and progression toward CKD (e.g. Cantaluppi V et al, Kidney Int, 2012).
We agree with Reviewer 3 that EPC-derived EVs have also a very interesting therapeutic potential in the treatment of renal diseases. We now have added a new paragraph in the revised manuscript. (page 17, lines 503-514).
“Circulating Endothelial progenitor cells (EPC) play an important role in the maintenance of vascular integrity and the regeneration of the vascular system. Circulating EPC number in CKD patients is 30% lower than in healthy controls [177-179]. Surdaki et al. showed that EPC number is in fact inversely correlated to the degree of kidney dysfunction [180]. The low EPC count [181] and dysfunction [182] may contribute to increase risk of CVD in CKD patients [183,184]. EPC therapy has been shown to decrease inflammation, proteinuria and preserve kidney function [185]. This beneficial effect is thought to be mediated in part in a paracrine manner and recent studies showed that EPC-EV may be responsible of this effect by transferring miRNA. Thus, EPC-EVs activate angiogenesis by transfer of mRNA, especially miR 126 and miR296 which have been shown to modulate proliferation, angiogenesis, apoptosis and inflammation [186-188]. EPC-EV also protect against progression CKD after IRI by inhibiting in particular tubule interstitial fibrosis [188].”